# Outlier Detection and Explanation Method Based on FOLOF Algorithm

**DOI:** 10.3390/e27060582

**Published:** 2025-05-30

**Authors:** Lei Bai, Jiasheng Wang, Yu Zhou

**Affiliations:** 1School of Electrical Engineering, North China University of Water Resources and Electric Power, Zhengzhou 450045, China; bailei@ncwu.edu.cn; 2School of Mechanical and Electrical Engineering, Wuhan University of Technology, Wuhan 430070, China; 3Engineering Training Center, North China University of Water Resources and Electric Power, Zhengzhou 450045, China; wangjiasheng@ncwu.edu.cn

**Keywords:** outlier detection, outlier analysis, objective function, golden section, prune, outlier factor

## Abstract

Outlier mining constitutes an essential aspect of modern data analytics, focusing on the identification and interpretation of anomalous observations. Conventional density-based local outlier detection methodologies frequently exhibit limitations due to their inherent lack of data preprocessing capabilities, consequently demonstrating degraded performance when applied to novel or heterogeneous datasets. Moreover, the computation of the outlier factor for each sample in these algorithms results in considerably higher computational cost, especially in the case of large datasets. This paper introduces a local outlier detection method named FOLOF (FCM Objective Function-based LOF) through an examination of existing algorithms. The approach starts by applying the elbow rule to determine the optimal number of clusters in the dataset. Subsequently, the FCM objective function is employed to prune the dataset to extract a candidate set of outliers. Finally, a weighted local outlier factor detection algorithm computes the degree of anomaly for each sample in the candidate set. For the analysis, the Golden Section method was used to classify the outliers. The underlying causes of these outliers can be revealed by exploring the anomalous properties of each outlier data point through the outlier factors of each dimension property. This approach has been validated on artificial datasets, the UCI dataset, and an NBA player dataset to demonstrate its effectiveness.

## 1. Introduction

Outliers are commonly defined as data points that deviate significantly from the majority in a dataset, generally indicating distinct underlying mechanisms [1]. They are classified into individual outliers and collective outliers based on quantity and nature [2]. Additionally, depending on the intrinsic characteristics of the data, outliers can be further categorized as vector outliers, sequence outliers, trajectory outliers, and graphical outliers [2]. The main goal of outlier detection is to pinpoint these outlier data points within a dataset, and this remains an area of current research. It is critical to highlight that outlier detection is widely applied in various fields, including fraud detection [3,4], network intrusion detection [5,6], public safety [7,8], image processing [9,10], non-destructive evaluation [11], and environmental monitoring [12]. In addition to outlier detection, outlier analysis is a crucial aspect of data mining. While outlier detection primarily aims to identify outliers in a dataset, outlier analysis delves into uncovering the underlying knowledge behind outliers. This involves classifying outliers to extract valuable insights or patterns. The systematic analysis and contextual interpretation of anomalous observations provides dual benefits: facilitating rigorous assessment of dataset integrity while enabling profound insights into the etiological factors, operational impacts, and intrinsic behavioral patterns characterizing these deviations.

In this paper, a data-driven approach that is based on the k-means clustering and local outlier factor (LOF) algorithm has been proposed and deployed for the management of non-destructive evaluation (NDE) in a welded joint. The k-means clustering and LOF model algorithm, which was implemented for the classification, identification, and determination of data clusters and defect location in the welded joint datasets, were trained and validated such that three different clusters and noise points were obtained.

Detection methods for outliers commonly fall into four major categories: statistics-based, distance-based, density-based, and clustering-based methods [2,13]. Beyond outlier detection, outlier explanation constitutes a crucial aspect of outlier mining [14]. While detection primarily involves the precise identification and isolation of outliers within a dataset, analysis extends to the methodical exploration of the underlying knowledge embedded within the detected outliers. This includes the systematic classification of outliers, enabling the derivation of valuable scientific conclusions or generalizable patterns. It is empirically observed that analysis of outliers generally yields more substantive and actionable knowledge compared to regular data analysis. The ultimate purpose of interpreting outlier data is to assist users in objectively evaluating the overall data quality, as well as to gain a deeper scientific understanding of the root causes, system-level effects, and distinct behavioral manifestations associated with outlier data. In recent research, several emerging explanation methods have provided novel insights for outlier detection, including LIME-AD [15], DTOR [16], and ProtoPNets [17].

In this paper, we propose an outlier detection and explanation method based on the FOLOF algorithm. The main contributions of this paper are divided into two aspects: outlier detection and explanation.

Regarding outlier detection, the process involves three main steps: (1) Preprocessing the dataset, utilizing the elbow rule to determine the optimal number of clusters for enhanced results due to the involvement of the FCM clustering algorithm. (2) Determining the objective function of FCM based on the sum of distances between all samples and cluster centers in the dataset. Removing one sample inevitably reduces the objective function value, but removing samples distant from the cluster center results in a more significant reduction. The reduction in objective function values is used to assess the outlier status of a sample, resulting in the removal of most outlier samples to create an outlier candidate set. (3) Using information entropy to represent the degree of discreteness in each dimension’s features. The LOF method is then applied to obtain outlier scores for each dimension in the outlier candidate set. Finally, a weighted approach is used to derive the final outlier score for each sample, with the scores arranged in descending order.

Regarding outlier explanation, the process involves three main steps: (1) Utilizing the FOLOF detection algorithm to identify the Top p outliers. Throughout the detection process, the corresponding outlier factor is recorded for each identified outlier. (2) Categorizing outliers into n levels (T_1_, T_2_, …, T_n_) based on the sizes of the outlier factors and employing the Golden Section method. Each dimension attribute is assigned an outlier level, and the degree of the outlier is positively correlated with the assigned level. (3) Experimental results validate the effectiveness of this method in revealing the underlying knowledge behind the outliers.

The remainder of this paper is organized based on the following sections. Section 2 looks at the related works. Section 3 describes the proposed method. Section 4 describes the improved outlier detection method FOLOF. Section 5 uses the FOLOF algorithm to interpret the outliers. Section 6 gives a detailed presentation of the experiment setting and discusses the results. Finally, we conclude the whole paper and outline possible future works in Section 7.

## 2. Related Work

In the field of outlier detection, distance-based outlier detection methods are common and straightforward. Ramaswamy et al. [18] introduced the k-nearest neighbor (KNN) outlier detection method based on the distance between samples and their neighbors. In this approach, the distance between a sample and its k-th nearest neighbor is used as a measure of sample outliers. A larger distance indicates a higher degree of outliers. Yang et al. [19] proposed the mean-shift outlier detector (MOD) method, which involves mean-shift processing of the dataset by computing the average of the nearest neighborhood of each sample. After three iterations, the dataset is modified by mean-shift, and sample outliers are determined by measuring the shift distance before and after mean-shift processing. However, the above methods do not consider the relationship between samples and their k-nearest neighbors.

In order to address uneven density between clusters, as well as to accommodate more complex data structures, Breunig et al. [20] introduced the concept of local outlier factor (LOF). This approach assigns an outlier factor to each sample based on its local density, providing a quantitative measure to assess the deviation of a sample from its k-nearest neighbors. The LOF method achieves accurate outlier detection even in the absence of a precisely defined sample distribution. To further enhance the LOF algorithm, Tang et al. [21] proposed the connectivity-based outlier factor (COF) method, which modifies the neighborhood computation of LOF to incremental computation and utilizes link distance to improve outlier detection performance. Similarly, Latecki et al. [22] presented the local density factor (LDF) method based on LOF. LDF estimates the local density of a sample’s neighborhood and determines the sample’s outliers by computing the ratio between the local density of the sample and that of its neighbors. However, a drawback of this approach is its substantial computational overhead, as it necessitates sequentially determining the neighborhood of each sample and subsequently computing its local outlier factor. In consideration of improving the LOF algorithm, Wang et al. [23] initially employed the DBSCAN clustering algorithm to trim the dataset, obtaining an initial set of outlier samples. Subsequently, the LOF algorithm is applied to evaluate the local outliers of objects within this set. This approach enhances the outlier detection accuracy and reduces the time required for outlier detection. Nonetheless, it is worth noting that the DBSCAN clustering algorithm presents challenges in parameter tuning, especially when dealing with unknown datasets. Cheng et al. [24] employ isolation forest and local outlier factor (LOF) for outlier detection. This approach first employs the iForest algorithm to rapidly scan the dataset, filter out normal data points, and generate a candidate set of potential outliers. Subsequently, based on the calculated outlier coefficients, we develop an adaptive threshold pruning method determined by the data’s outlier degree. Finally, the LOF algorithm is implemented to further refine the candidate outlier set, obtaining more precise and reliable outlier scores.

In the field of outlier explanation, Knorr and Ng [25] classified distance-based outliers into two categories: trivial and nontrivial, further distinguishing nontrivial outliers into strong and weak based on their different features. They analyzed the outlier characteristics of the data from attribute combinations at different levels, starting from the lowest level. Building upon this, Chen et al. [26] proposed a related search algorithm based on the previous method. They argue that different attribute subspaces influence the formation of outliers, and divide the attribute space into an identification subspace, an indication subspace, and an observation subspace. However, it is worth noting that this algorithm requires some domain knowledge when choosing parameters, which may lead to different results for different observations. In [27], the concept of “outlier attribute” and “outlier cluster” was introduced, and an outlier analysis method based on outlier classification was presented. This approach involves the development of an outlier classification approach using clustering techniques, which serves to uncover the underlying knowledge hidden within the outliers.

In the latest research, Bhurtyal S et al. [28] present a deep learning architecture exploiting a feature choice module and a mask generation module in order to learn both components of explanations. The extensive experimental campaign carried out on synthetic and real datasets provides empirical evidence demonstrating the quality of the results returned by M^2^OE for explaining single outliers and M^2^OE-groups for explaining outlier groups. Papastefanopoulos V et al. [29] introduce an interpretable approach to unsupervised outlier detection by combining normalizing flows and decision trees. Post hoc statistical significance testing demonstrated that interpretability in unsupervised outlier detection can be achieved without significantly compromising performance, making it a valuable option for applications that require transparent and understandable anomaly detection. Angiulli F et al. [30] presents a deep learning architecture exploiting a feature choice module and a mask generation module in order to learn both components of explanations. Experiments were conducted on both artificial and real datasets, and compared with competitors to validate the effectiveness of the proposed method.

Although deep learning has manifested remarkable advantages in the representation of high-dimensional nonlinear data, its black-box characteristic might lead to difficulty in effectively decoupling the correlation between detection results and underlying features. The merits of traditional algorithms based on clustering and the Local Outlier Factor (LOF) algorithm lie in being capable of conducting rapid outlier detection on low-dimensional datasets and fulfilling the scene requirements of high interpretability through the feature differences among the data.

## 3. Methods

This Section systematically elaborates the theoretical foundations of the proposed methodology through three subsections. Section 3.1 introduces the Local Outlier Factor (LOF) algorithm for outlier detection. Section 3.2 details the Elbow rule employed to determine the optimal number of cluster centers in Fuzzy C-Means clustering. Finally, Section 3.3 presents the Golden Section for interpreting detected outliers.

### 3.1. Local Outlier Detection Algorithm LOF

The LOF algorithm quantitatively measures the outlier degree of each sample object using the outlier factor. Instead of computing the global density of samples, it computes the local reachability density of a sample by considering its k-nearest neighbors. Conventional LOF algorithms require explicit definitions of the following:

(1) d (p, o): the distance between point p and point o.

(2) k-distance: the k-distance of p is the distance from p to the kth distant point around it, but does not include p.

(3) k-distance neighborhood of p: all points whose distance from point p does not exceed its kth distance, including points on the kth distance. Their point set is the kth distance neighborhood N_k_ (p) of p. Therefore, the number of points in the kth neighborhood of p and N_k_ (p) ≥ k.

(4) Reach distance: the kth reach distance from point o to point p is(1)$$reach-distance_{k}(p,o)=max\{k-distance(o),d(p,o)\}$$

The kth reach distance from point o to point p is the maximum of the kth distance of o and the true distance between p and o.

(5) Local reachability density: the local reachability density of point p is formally expressed as follows:(2)$$lrd_{k}(p,o)=1/(\frac{\sum_{O\in N_{k}(p)}reach-dist_{k}(p,o)}{\left|N_{k}(p)\right|})$$

(6) Local outlier factor: the local outlier factor of point p is formally expressed as follows:(3)$$LOF_{k}(p)=\frac{\sum_{O\in N_{k}(p)}\frac{lrd(o)}{lrd(p)}}{\left|N_{k}(p)\right|}=\frac{\sum_{O\in N_{k}(p)}lrd_{k}(o)}{\left|N_{k}(p)\right|}/lrd_{k}(p)$$

From Equation (3), it can be deduced that the local outlier factor of point p is determined by the ratio of its k-nearest neighbor’s local reachability density to the local reachability density of point p. If the local reachability density of point p is relatively tiny, while the local reachability densities of its k-nearest neighbors are relatively large, then the LOF of point p will be large. In turn, when the opposite condition applies, the LOF will be tiny. The proof goes as follows.

**Proof.** Assuming that the neighborhoods of the two points $p_{1}$ and $p_{2}$ are the same, as shown in Equation (4), and the local reachability density of $p_{1}$ is less than that of $p_{2}$, as shown in Equation (5).(4)$$\sum_{O\in N_{k}(p_{1})}lrd_{k}(o)=\sum_{O\in N_{k}(p_{2})}lrd_{k}(o)$$(5)$$lrd_{k}(p_{1},o)<lrd_{k}(p_{2},o)$$Combining Equations (3)–(5), we can derive the following:(6)$$LOF_{k}(p_{1})>LOF_{k}(p_{2})$$From Equation (6), it can be concluded that the degree of deviation of $p_{1}$ from this neighborhood is greater than that of $p_{2}$ from this neighborhood, so the outlier factor of $p_{1}$ is larger.                                                                 □

### 3.2. Elbow Rule

The FCM clustering algorithm uses an iterative classification approach. It starts by randomly generating c cluster centers and then iteratively updates the partition matrix and cluster center matrix until the FCM cost function reaches its minimum. As c increases, the number of samples within each cluster decreases, bringing them closer to their respective cluster centers and resulting in a reduction in the cost function. However, with additional increases in c, the rate of decrease in the cost function decreases until the curve levels off. Throughout the increment of c, the elbow represents the phase of maximum cost function reduction. Determining the value of c at this elbow point yields a valid clustering result. The cost function for FCM is expressed as shown in Equation (7).(7)$$cost(c)=\frac{\sum_{i=1}^{c}\sqrt{\sum_{j=1}^{N(i)}u_{ij}{\left(x_{ij}-C_{i}\right)}^{2}}}{\left|\sum_{i=1}^{c}N(i)\right|}$$
where $x_{ij}$ denotes the j-th sample of the i-th cluster, and $u_{ij}$ represent its corresponding degree of membership, *c* represents the number of clusters, *N*(*i*) denotes the number of samples in the i-th cluster, and C*_i_* represents the *i*-th cluster center.

### 3.3. Golden Section

The Golden Section method, also known as the best-section method, involves splitting a whole into two parts. The ratio of the larger part to the whole is equal to the ratio of the smaller part to the larger part, approximately 0.618. Widely recognized as the most aesthetically pleasing ratio, this proportion is referred to as the Golden Section [31,32]. Due to its strict proportionality and aesthetic appeal, the Golden Section finds broad applications in aesthetics, music, architecture, and transportation [33,34]. As shown in Figure 1, the length of the segment AC is denoted as a, and point B is located near point C in the Golden Section, and the length of the segment AB is denoted as b. The ratio of b to a is termed the golden ratio. Its length adheres to the following formula.

First, based on the formal definition of the golden ratio, as shown in Equation (8).(8)$$\frac{AB}{AC}=\frac{BC}{AB}$$

Substitute AB = b, AC = a, and BC = a − b into Equation (8), and simplify the expression in accordance with Equations (9)–(11).(9)$$b^{2}=a(a-b)=a^{2}-ab$$(10)$$a^{2}-ab+\frac{1}{4}b^{2}=\frac{5}{4}b^{2}$$(11)$$a-\frac{b}{2}=\frac{\sqrt{5}}{2}b$$

Simplifying Equation (11), we have(12)$$\frac{a}{b}=\frac{\sqrt{5}+1}{2}$$

Assuming the length of AC to be 1, point B is positioned at approximately 0.618, indicating that B is the optimal segmentation point in the interval AC.

## 4. Local Outlier Detection Method Based on Objective Function

### 4.1. Pruning Algorithm Based on the Objective Function of FCM

Leveraging the FCM objective function from reference [35], as shown in Equation (13), we introduce a novel pruning approach. Initially, the FCM algorithm is employed on the dataset to generate multiple clusters. Clusters with fewer members than the average cluster size are identified as small clusters, and their samples are included in the outlier candidate set. As per Equation (13), eliminating a sample from the dataset inevitably leads to a decrease in the objective function value. The reduction in the objective function value becomes more substantial if the removed sample is significantly distant from the cluster center, indicating a higher degree of outlier. This concept is used to assess the degree of outliers in the sample.(13)$$J(u,c)=\sum_{i=1}^{c}\sum_{j=1}^{n}u_{ij}^{m}{\left\|x_{j}-V_{i}\right\|}^{2}$$

Derivation process of the pruning algorithm based on the FCM objective function:

In Equation (13), let $x_{j}-V_{i}=d_{ij}$ represent the distance from the jth point to the ith cluster center, and n represents the number of samples in dataset D. When a point is removed from dataset D, the number of samples becomes n − 1, and the elimination of a point minimally affects the overall dataset’s cluster center. At this juncture, the FCM objective function transforms into Equation (14).(14)$$J_{0}(u,c)=\sum_{i=1}^{c}\sum_{j=1}^{n-1}u_{ij}^{m}d_{ij}^{2}$$

Clearly, in comparison to Equation (13), the removal of points from the dataset results in $J_{0}(u,c)<J(u,c)$, indicates a reduction in the objective function value. While clustering may misclassify some outliers as normal clusters, these outliers are typically located at the periphery of the cluster, remote from the cluster center. Eliminating such outliers increases the value of $\Delta \sum d_{ij}$, where $\Delta \sum d_{ij}(outlier)$ denotes the shift in distance caused by the removal of an outlier point, and $\Delta \sum d_{ij}(normal)$ denotes the shift in distance caused by the removal of a normal point. It can be readily deduced that $\Delta \sum d_{ij}(outlier)>\Delta \sum d_{ij}(normal)$.

Firstly, Equations (15) and (16) are utilized to represent the values of the objective function after pruning outlier and normal, respectively.(15)$$J_{1}(u,c)=\sum_{i=1}^{c}\sum_{j=1}^{n-1}u_{ij}^{m}d_{ij}^{2}(outlier)$$(16)$$J_{2}(u,c)=\sum_{i=1}^{c}\sum_{j=1}^{n-1}u_{ij}^{m}d_{ij}^{2}(normal)$$

As demonstrated in Equation (17), the distance between the outlier and the cluster center is greater than the distance between a normal and the cluster center.(17)$$\Delta \sum d_{ij}(outlier)>\Delta \sum d_{ij}(normal)$$

Therefore(18)$$J_{1}(u,c)<J_{2}(u,c)$$

Then, the values of *J*_1_ and *J*_2_ are respectively subtracted from the total objective function value *J* of the dataset to derive Equation (19).(19)$$J-J_{1}(u,c)>J-J_{2}(u,c)$$

Let(20)$$J-J_{1}(u,c)=\Delta J(outlier),J-J_{2}(u,c)=\Delta J(normal)$$

Eventually, Equation (21) is derived by integrating Equations (19) and (20).(21)ΔJ(outlier)>ΔJ(normal)

In other words, when the removed points exhibit significant outliers, the reduction in the objective function will be significantly larger compared to the reduction in the normal data. This criterion is applied to identify outliers in the dataset. The following parameters are introduced to elucidate the above approach:

(1) OF: the objective function value of the complete dataset.

(2) $OF_{i}$: the objective function value of the dataset after removing the ith point.

(3) $DOF_{i}=OF-OF_{i}$: reduction in objective function value after removing the ith point.

(4) $AvgDOF=\sum DOF_{i}/n$: average reduction after removing all points one at a time.

Introduce the threshold T, if DOFi>T(AvgDOF), it indicates that the outlier degree of the ith point is relatively high, and it is added to the outlier candidate set.
**Algorithm 1** Pruning algorithm based on objective function valueInput: dataset DOutput: outlier candidate set D.Step 1: The FCM algorithm is applied to the entire dataset to derive the OF and to include small clusters in the set of outlier candidates.Step 2: The remaining points are removed one by one to obtain all OF_i_, and each DOF_i_ and average reduction AvgDOF_i_ are calculated.Step 3: Compare DOF_i_ and T(AvgDOF). If DOFi>T(AvgDOF), put the ith point into the outlier candidate set.

### 4.2. Weighted LOF Algorithm

Information entropy is commonly used to characterize the dispersion of data. The greater the entropy, the greater the dispersion of variables and the less information it provides. The smaller the entropy value, the smaller the degree of discreteness of the variable and the additional information it provides. Therefore, the information provided by the features of different dimensions of the data is different, and different weights need to be given accordingly. For properties with large entropy, the weights should be higher, so that the degree of dispersion is more obviously reflected.
**Algorithm 2** Weighted LOF algorithmInput: dataset D;Output: Weight Set W={ω1,ω2,…,ωm}, outlier value score of each data LOFi;Step 1: Perform Z-score standardization on dataset D to obtain D′, and calculate the proportion P_ij_ of the jth dimension attribute of the ith data in dataset D′ through Equation (22).Pij=xij′∑i=1nxij′ (22)where 1 ≤ i ≤ n, 1 ≤ j ≤ m, x′ij is the jth dimensional attribute value of the ith data of D′.Step 2: Calculate the information entropy E_j_ of the jth dimension attribute in dataset D′ with Equation (23).Ej=−p∑i=1nPijlnPij (23)where p=1/lnn.Step 3: Calculate the weight ω_j_ of the jth dimensional attribute with Equation (24)ωj=Ej∑j=1mEj (24)After the above steps, m dimension attribute weight sets are finally obtained, W={ω1,ω2,…,ωm}.where 0 ≤ ω_j_ ≤ 1 and ∑j=1mωj=1.Step 4: After each one-dimensional attribute is weighted by the entropy weight method, the traditional LOF algorithm is executed separately for each dimensional attribute, and it is weighted according to Equation (25) to obtain the final LOF value as the anomaly score of the overall data.LOFi=∑j=1mωjLOFij (25)where LOF_ij_ denotes the LOF value of the jth dimension of the ith data.

### 4.3. Description of an Objective Function-Based Local Outlier Detection Algorithm

Combining Algorithm 1 in Section 4.1 and Algorithm 2 in Section 4.2, we can obtain an outlier detection method based on FOLOF, as explained in Algorithm 3. The outlier detection performance of conventional LOF algorithms is not ideal for unknown datasets, where the data distribution and the number of clusters are not known. This is because computing the degree of outliers for each data point significantly increases the computational complexity of the algorithm. The FOLOF algorithm makes improvements in this respect.
**Algorithm 3** Outlier detection method based on the FOLOF algorithmInput: dataset D, number of neighborhood queries k, pruning threshold T, number of outliers p;Output: Top p outliers;Step 1: The elbow rule from Section 3.2 is used to determine the optimal number of clusters c for dataset D, to obtain the best clustering effect.Step 2: Set the parameter c, apply the FCM algorithm to the cluster data D, and record and compare the change in objective function values upon removal of each data point. Prune dataset D based on the pruning threshold T(AvgDOF_i_) to derive the outlier candidate set D_0_.Step 3: In the outlier candidate D_0_, each dimension property is weighted by Equations (22)–(24), and then the outlier score LOF of each data is calculated by Equation (25).Step 4: Arrange the LOF values of each data in order from largest to smallest, output the first p data objects and form the outlier set of dataset D_0_.

## 5. Outlier Explanation Method Based on FOLOF Algorithm

Based on the analysis in Section 3.3, this paper proposes an outlier factor classification method based on the Golden Section method, which can accurately mine the latent knowledge behind the outliers and find the causes of the outliers. The overall outlier factor for each data point in the outlier candidate set and the individual outlier factors for the features in each dimension can be obtained by the FOLOF algorithm. The size of the outlier factor, that is, the LOF value, indicates the degree of outlier of the data. Based on its range of values, the degree of outliers can be divided into different levels by the Golden Section method. Since the number of outliers in the lower level is generally larger than the number of outliers in the higher level, the idea of the Golden Section is perfectly reasonable.

It is assumed that the value range of the outlier factor of the data is [a, b]. It is split into n distinct levels. First, the whole area is divided according to the Golden Section, then the remaining area is divided according to the Golden Section, and the corresponding levels are defined as $T_{1},T_{2},\ldots ,T_{n}$, whose upper limits are defined as $t_{1},t_{2},\ldots ,t_{n}$, and after induction and classification, we have$$t_{1}=(1-0.618)a+[{\left(1-0.618\right)}^{0}]0.618b$$$$t_{2}={\left(1-0.618\right)}^{2}a+[{\left(1-0.618\right)}^{1}+{\left(1-0.618\right)}^{0}]0.618b$$$$t_{i}={\left(1-0.618\right)}^{i}a+[{\left(1-0.618\right)}^{i-1}+{\left(1-0.618\right)}^{i-2}+\ldots +{\left(1-0.618\right)}^{1}+{\left(1-0.618\right)}^{0}]0.618b$$

The range of values for each outlier level can be obtained in turn. Suppose that n takes the value 3, which can be divided into three classes: weak, ordinary, and strong outliers. The various outliers are defined as follows.

**Definition** **1.**
*Weak outlier: Outliers at T_1_ level, and the value of outlier factor LOF(T_1_) conforms to the following formula:*

$$LOF_{\min}\leq LOF(T_{1})<(1-0.618)LOF_{\min}+0.618LOF_{\max}$$



**Definition** **2.**
*Ordinary outlier: Outliers at T_2_ level, and the value of outlier factor LOF(T_2_) conforms to the following formula:*

$$(1-0.618)LOF_{\min}+0.618LOF_{\max}\leq LOF(T_{2})<(1-0{.618)}^{2}LOF_{\min}+[{\left(1-0.618\right)}^{1}+{\left(1-0.618\right)}^{0}]0.618LOF_{\max}$$



**Definition** **3.**
*Strong outlier: Outliers at T_3_ level, and the value of outlier factor LOF(T_3_) conforms to the following formula:*

$${(1-0.618)}^{2}LOF_{\min}+[{\left(1-0.618\right)}^{1}+{\left(1-0.618\right)}^{0}]0.618LOF_{\max}\leq LOF(T_{3})<LOF_{\max}$$



The application of the Golden Section method to the classification of outliers is illustrated by the introduction of iso-outlier factor lines.

As shown in Figure 2, the center of the circle represents the minimum value a of the outlier factor, the outermost circumference represents the maximum value b of the outlier factor, and the same circumference represents the same outlier factor. Taking the length of this range as the radius, it is divided into T_1_, T_2_, and T_3_ according to the Golden Section method described above, representing the three outlier levels, with the outlier factor gradually increasing along the dashed line direction. In addition, there are numerous cases of outlier distribution in real datasets, as shown in Figure 3.

In Figure 3a, there is only one cluster, and the red point is the outlier. The further away the cluster is, the larger the outlier factor, that is, the higher the outlier level. The blue dot in Figure 3b is the k-neighborhood of the corresponding outlier. The points around the k-neighborhood of the outlier at level T_1_ are relatively sparse; that is, if their k-neighborhood density is small, then the outlier factor of the point is small, and the outlier level is low. Points around k-neighborhoods of outlier level T_2_ are relatively dense; that is, if the neighborhood density of k-neighborhoods is large, then the outlier factor of a point is large, and the outlier level is extreme.

## 6. Experimental Analysis

In this part, the feasibility of the proposed algorithm will be verified from the aspects of clustering effect (CH index, Dunn index, I index, S index) [36,37], accuracy, false detection rate [38,39,40], and so on, and the causes of corresponding outliers will be explained to excavate the connotation knowledge behind it. All the parameters mentioned in the previous Section are set uniformly in the following experiments. The threshold T, used in pruning, is one, and the number of neighborhoods k in the LOF and FOLOF algorithms is five. The algorithm is written in MATLAB R2020a, and the experimental environment is an AMD R7 3.2 GHz CPU, 8.00 GB RAM, and Windows 11 operating system.

### 6.1. Synthetic Dataset

#### 6.1.1. Outlier Detection in Synthetic Dataset

For an intuitive demonstration of the entire process of outlier detection and interpretation, we chose to experiment with the synthetic dataset D1. D1 contains a total of 70 data points, including 10 outliers located at data points 6, 7, 19, 30, 40, 45, 52, 58, 69, and 70. For visualization of the synthetic dataset, Figure 4 is provided, where the outliers are shown in red and the normal data points in black.

As shown in Figure 5, the cost function gradually tends to flatten as the number of clusters changes from 3 to 4, so the optimal number of clusters c is 3. Running the pruning algorithm obtains the value of OF, 625.587, and the value of Avg (LOF_i_), 9.3017. Each DOF_i_ is listed in Table 1.

Figure 6 shows a plot of the objective function value after removing the data point, and Figure 7 shows a needle plot of the objective function value as a function of the data point. Compared with Avg (LOF_i_), 11 points, such as 6, 7, 19, 30, 40, 45, 52, 58, 60, 69, and 70, are obtained by pruning, which are put into the outlier candidate set. The candidate set contains all outliers with a pruning accuracy of 100 percent.

The weighted LOF algorithm is run to output the top 10 points with the largest outliers. The outlier factors for each point and each one-dimensional outlier are listed in Table 2, including 10 points such as 58, 69, 60, 6, 40, 19, 7, 52, 30, and 45, of which 9 are true outliers. After removing the detected outliers and reclustering the dataset, the clustering effect is improved, as shown in Table 3.

To evaluate the performance of outlier detection, two metrics, Precision and Noise factor, are used, denoted by Pr and Nf, respectively. TP is used to denote the number of true outliers detected by the algorithm, and FP is used to denote the number of normal data points that the algorithm incorrectly detects as outliers.(22)$$Pr=\frac{TP}{TP+FP}\;Nf=\frac{FP}{TP+FP}$$

Pr and Nf have a maximum of one and a minimum of zero, respectively. The larger the value of Pr, the smaller the value of Nf, and the better the performance of the algorithm. For the detection results of dataset D1, TP = 9 and FP = 1, and Pr = 0.9 and Nf = 0.1 are calculated, and the results are extremely excellent.

#### 6.1.2. Explanation of the Outliers in the D1 Dataset

Compared to the manually set outliers, the outlier factor of the 60th point detected by the algorithm is considerably larger than that of the 70th point, and thus it is ranked outside the top 10 outliers. By looking at the contents of Table 1, it can be concluded that the outlier factors of the 11 points in the first dimension are not very different. The reason why the outlier factor of the 60th point is larger than that of the 70th point is mainly in the second dimension, where the outlier factor of the 60th point is much larger than that of the 70th point.

For outlier factors: Weak outliers = {6, 7, 19, 30, 40, 45, 52, 70}, Ordinary outliers = {60}, and Strong outliers = {58,69}. The classification results are shown in Figure 8.

Similarly, for the first-dimensional outlier factor: Weak outliers = {7, 19, 30, 45, 52, 58, 60, 69, 70}, Ordinary outliers = {40}, and Strong outliers = {6}. The classification results are shown in Figure 9.

For the second outlier factor: Weak outliers = {6,7,19,30,40,45,45,52,70}, Ordinary outliers = {60}, and Strong outliers = {58,69}. The classification results are shown in Figure 10.

As can be seen from the classification results, the 70th point belongs to the weak outliers in both dimensions, while the 60th point belongs to the weak outliers in the first dimension, but to the ordinary outliers in the second dimension. Thus, in general, the degree of outliers at the 60th point is much higher than at the 70th point. In addition, in the classification of outlier factors, the 6th point belongs to weak outlier, while in the first-dimensional outlier factor, the 6th point belongs to strong outlier; thus, the first-dimensional outlier factor is the main reason for the 6th point outlier, which directly leads to its elevated degree of outlier.

According to the analysis in Section 3.1, the direct cause of the size of the outlier factor is the achievable density, namely, the lrd_k_-value. The reachable densities for the 11 points in the candidate set are listed in Table 4, where the specific causes of the data outliers can be clearly observed.

By observing the contents of the table, we can see that the reachable density of 11 points in the first dimension is not much different, while the reachable density of the 60th point in the second dimension is greatly inferior to that of the 70th point, which leads to a large outlier factor. Therefore, the cause of the 60th point outlier is mainly in the second dimension, which is consistent with the analysis of the outlier factor.

### 6.2. UCI Datasets

#### 6.2.1. Outlier Detection in UCI Datasets

In this Section, the performance of the proposed algorithm is verified by comparing the results of its operation with the LOF algorithm on real-world datasets. The UCI dataset parameters are shown in Table 5.

In the Iris dataset, the first two data types are selected as normal data, and the 10 data points in the third category are selected as outlier data. Similarly, a total of 130 data points from the first two categories of the Wine dataset are selected as normal data, and the third category is chosen randomly. Eight data points are considered as outliers. For the Yeast dataset, it is not necessary to use all the data. Three major categories, such as CYT, NUC, and MIT, were selected as normal data with a total of 1136 data points, and 20 data points in the POX category were selected as outliers. Moreover, due to its dimension 6, this property contains zero values. When running the attribute weighting algorithm to compute the ratio, the presence of the zero value is not allowed, and the dimension 6 attribute is removed, and the remaining dimension 7 attribute is used for outlier detection. Similarly, for the UKM dataset, since extremely low contains 0 values, a total of 224 data points in High and Middle are selected as normal data, and 5 data points in Low are randomly selected as outlier data. For the Seeds dataset, the first and third classes are selected as normal data, and 10 data points from the second class are chosen as outliers. For the Wdbc dataset, the first and second classes are taken as normal data, and five data points are randomly generated as outliers. For the Speech dataset, 10 data points are randomly generated as outliers.

The pruning accuracy, that is, the ratio of the actual number of outliers in the candidate outliers obtained after pruning to the number of outliers in the entire dataset, and the amount of remaining data after pruning, proves that the FCM pruning method is better than the DBSCAN [24] and PMLDOF [41] pruning method, as shown in Table 6 and Table 7.

By comparing the detection results with the clustering effect after removing the detected outliers, we demonstrate the superiority of the proposed method over the LOF algorithm. The experimental results are shown in Table 8 and Table 9. The running time is t in units of s.

Simulation experiments show that the proposed algorithm has some processing power for unknown datasets and significantly improves detection results compared to traditional LOF algorithms. After removing the detected outliers, the dataset is again clustered, and the clustering effect is significantly improved. Before detection, the data are pruned through FCM, which reduces a large fraction of the computational effort. Compared to the LOF algorithm, the running time is also significantly reduced.

To further evaluate the detection efficacy of FOLOF, we conducted comparative experiments on UCI datasets, comparing it against four state-of-the-art detectors: LOF [17], IFOREST [42], BLDOD [43], and PEHS [44]. The comparative analysis employed Area Under the ROC Curve (AUC) as the performance metric, with detailed experimental outcomes systematically documented in Table 10. Experimental results demonstrate that FOLOF outperforms state-of-the-art contrastive methods in AUC metrics, highlighting its enhanced discriminative capability and reliability in comparative evaluations.

To rigorously analyze experimental outcomes and evaluate performance disparities between our proposed method and comparative approaches, we performed Wilcoxon rank-sum tests on AUC metrics across UCI datasets. This statistical method quantifies inter-method differences while revealing directional trends through positive and negative rank-sum indicators (R+/R−). When the *p*-value fell below the predefined significance threshold (α = 0.05), it confirmed statistically significant differences between method pairs, with R+/R− relationships explicitly indicating performance superiority. As demonstrated in Table 11, our FOLOF method exhibited statistically significant advantages over all baseline algorithms in the UCI benchmark evaluations.

#### 6.2.2. Explanation of the Outliers in the UCI Datasets

After completing the outlier detection, we analyzed the top n outliers in each UCI dataset. Using the Golden Section method, these outliers were classified into weak outliers, ordinary outliers, and strong outliers. The specific classification results are shown in Table 12.

By analyzing the classification results, it is evident that the first-dimensional feature, the sepal length, is one of the main reasons for the outlier status of point 107 in the Iris dataset, which has a relatively short sepal length. Similarly, in the Wine dataset, the third-dimensional feature Ash is one of the main reasons for the outliers at points 135, 136, and 139, where the Ash values are all relatively small. For the Yeast dataset, the fifth-dimensional feature, ERL, is one of the main causes of the outliers at points 1123, 1129, 1130, 1131, and 1132, where 1123 and 1130 have correspondingly modest attribute values while the others are relatively large. In the UKM dataset, the second-dimensional feature SCG is one of the main reasons for the outlier status of point 229, which has a minor attribute value. By classifying outlier data, we can gain detailed insights into the specific causes of data outliers, and combining this with actual data allows for a deeper exploration of the features behind the data. In the Seeds dataset, the second dimension is the main cause of outliers; 141, 143, 145, 146, 149, and 150 are weak outliers, 142 is an ordinary outlier, and 147 and 148 are strong outliers. The Wpbc and Speech datasets have many dimensions, so we use the overall outlier factor as the basis for determining the type of outlier. In the Wpbc dataset, 572 and 573 are weak outliers, 574 is an ordinary outlier, and 570 is a strong outlier. In the Speech dataset, 3689, 3694, and 3695 are weak outliers, 3687, 3690, and 3696 are ordinary outliers, and 3691 and 3692 are strong outliers.

### 6.3. NBA Player Dataset

This Section aims to apply the outlier explanation method to categorize the data of all experienced, second-year, and rookie players in the 2020–2021 NBA season, revealing the underlying knowledge of the outlier players. The dataset consists of 150 data points covering five features: average points, average rebounds, free throws, offense, and defense.

Based on the FOLOF detection algorithm, the top 10 points with the largest outlier factors and the outlier factors in each dimension are detected, as shown in Table 13.

The ten players corresponding to the above points are Luka Doncic, De’Aaron Fox, Giannis Antetokounmpo, James Harden, Moses Brown, Brodic Thomas, LeBron James, Stephen Curry, Damian Lillard, and Darius Garland. Each dimensional outlier factor of these ten points is classified, and the players’ names are replaced with English abbreviations; for example, Luka Doncic is abbreviated as LD. Five representative players are selected for analysis, namely, LD (6), GA (5), JH (21), MB (79), and SC (1). The results of the classification based on the local outlier factor for each dimension of these 10 NBA players are shown in Table 14.

For SC (1), the first-dimension feature, that is, score, is the main reason for its deviation. Combined with the actual data, SC has a field-averaged score as high as 32, while its magnitude stars have field-averaged scores of about 25. As a superstar, SC is extremely good at scoring and is one of the best defenders today. SC is not particularly prominent among the other capacities. This makes his bias not as large as LD’s.

It is clear that for 6 and 21, the second-dimensional feature, the average rebounds per game, is the main reason for their deviation. Combined with the actual numbers, LD and JH averaged 8 and 7.9 rebounds per game in the series, while their stars at the same level averaged about 5. LD and JH have superior rebounding per game compared to other guards, which is perfectly consistent with reality.

For GA (5), the third-dimensional feature, namely, free throws, is the main reason for its deviation. Combined with the actual data, GA’s free throw hit rate in the series is only 68.5, while the free throw hit rate of its stars of the same level is about 85 or even higher, indicating that GA’s free throw is its biggest weakness, which is significantly lower than the NBA average level. He can work on his free throws more in future workouts.

For MB (79), the fourth-dimension feature, that is, the attack value, is the main reason for its outlier. Combined with the actual data, MB’s offensive rating is as high as 3.6, compared to about 1 for stars of the same rank, implying that MB has been particularly offensive efficient this season. He is the most prominent player in the two-year group and deserves to be the focal point of the team.

In the NBA, their general capacity is greatly higher than that of the average man, and it is typically difficult to exceed the limits of humanity. Thus, the weak outliers account for the majority in the classification. Few players reach the level of the ordinary outlier in a given ability, and for players with better statistics, players with strong outlier levels already represent the highest level of the NBA in terms of corresponding ability.

### 6.4. Analysis and Discussion

In this subsection, a thorough and comprehensive analysis of the proposed FOLOF algorithm is provided, focusing on five key aspects.

(1) The FOLOF algorithm determines the best number of clusters in the datasets through the elbow rule to achieve the best clustering effect, which makes it have better processing ability in the face of unknown datasets.

(2) The FOLOF algorithm prunes the dataset through the change in the objective function of FCM to obtain the preliminary outlier set, which considerably reduces the computational complexity in terms of the amount of data. However, the LOF algorithm does not pre-process the dataset, which leads to a large amount of computation and a high running time.

(3) In terms of input parameters, the FOLOF algorithm only needs to manually input the number of neighborhood queries k, pruning threshold T, and the number of outliers p to be output, and the setting of these parameters does not need to go through a cumbersome experimental process. In general, empirical values can be used to obtain excellent detection results.

(4) In the aspect of outlier explanation, the algorithm proposed in this paper can successfully mine outlier attributes according to the size of the outlier factor and through the Golden Section method, and classify each outlier, so as to obtain more useful knowledge and rules.

(5) Due to the inherent limitations of the FCM clustering algorithm when handling datasets with arbitrarily shaped clusters, particularly those with non-convex structures, its pruning capability is reduced compared to datasets with convex cluster structures. Consequently, it is necessary to retain a higher proportion of candidate samples in the detection set to ensure the accuracy of outlier identification.

## 7. Conclusions

To enhance outlier detection accuracy and explore the underlying knowledge, this paper proposes a novel outlier detection method. For outlier detection, we introduce an approach integrating FCM clustering and LOF algorithms. The method first determines optimal cluster numbers through the elbow rule, then applies the FCM clustering algorithm to partition datasets while pruning them via objective function optimization, and finally employs a weighted LOF algorithm to evaluate outlier scores. Experimental comparisons with four classical or advanced outlier detection methods on UCI datasets demonstrate that the FOLOF algorithm achieves significantly superior detection accuracy.

Regarding outlier interpretation, this study initially identifies top-p outliers, then records each outlier’s global outlier factor and dimensional-specific outlier factors. Through Golden-Section-based regional partitioning, we systematically derive root causes of anomalies based on dimensional outlier levels. The algorithm proposed in this paper can successfully mine outlier attributes according to the size of the outlier factor and through the Golden Section method, and classify each outlier, so as to obtain more useful knowledge and rules.

The FCM algorithm demonstrates effective clustering performance primarily for spherical data distributions, yet exhibits limitations when processing non-spherical or complex datasets. This inherent constraint of FCM consequently affects the proposed method’s capability in handling sophisticated datasets. Future work will concentrate on enhancing the algorithm’s adaptability to complex data patterns and developing an automated parameter optimization framework to address potential performance degradation caused by empirical parameter settings.

## Figures and Tables

**Figure 1 entropy-27-00582-f001:**
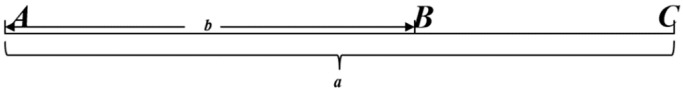
Golden Section method.

**Figure 2 entropy-27-00582-f002:**
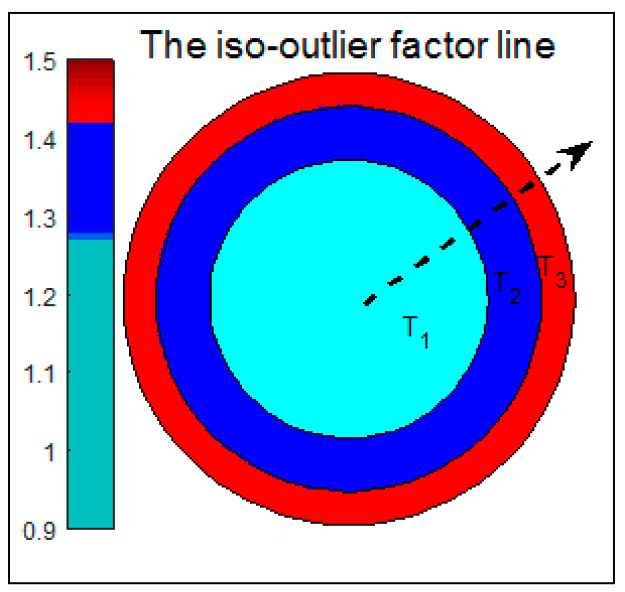
The iso-outlier factor line diagram.

**Figure 3 entropy-27-00582-f003:**
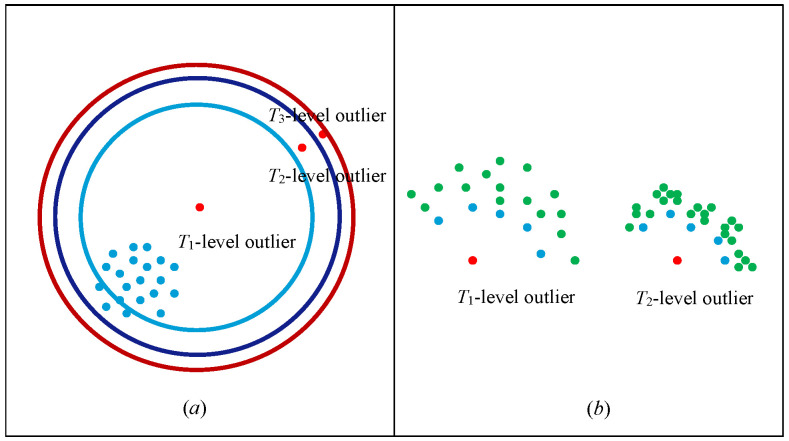
Schematic diagram of outlier classification. (**a**) Different outlier values indicate. (**b**) Schematic diagram of k-neighborhood.

**Figure 4 entropy-27-00582-f004:**
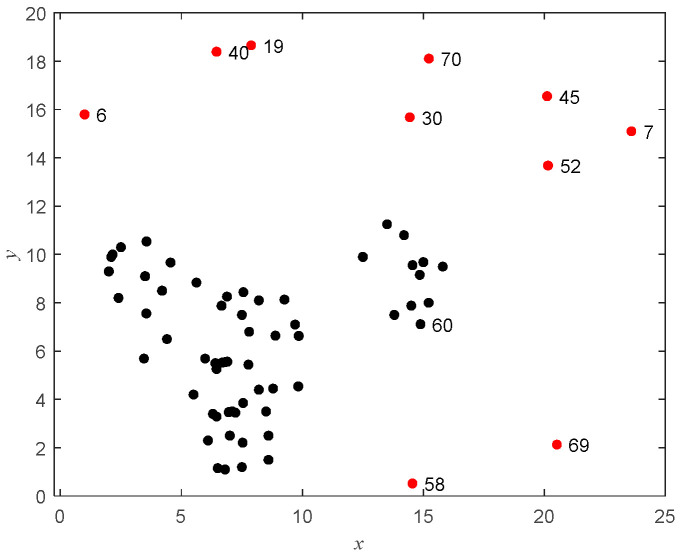
Synthetic dataset D1.

**Figure 5 entropy-27-00582-f005:**
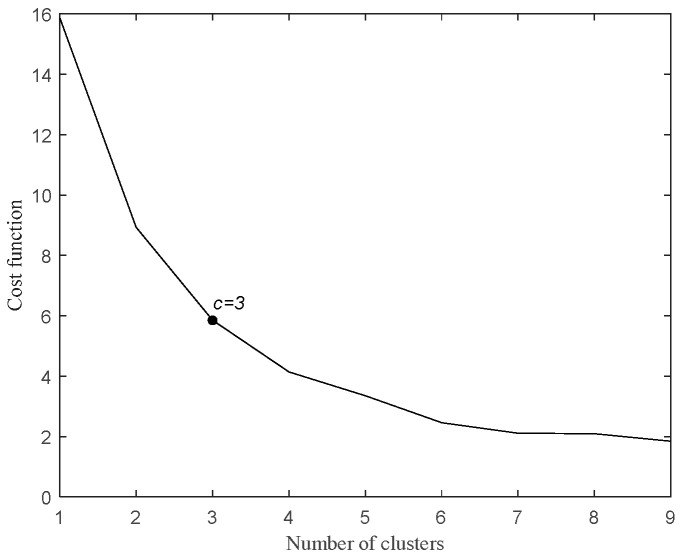
Elbow rule.

**Figure 6 entropy-27-00582-f006:**
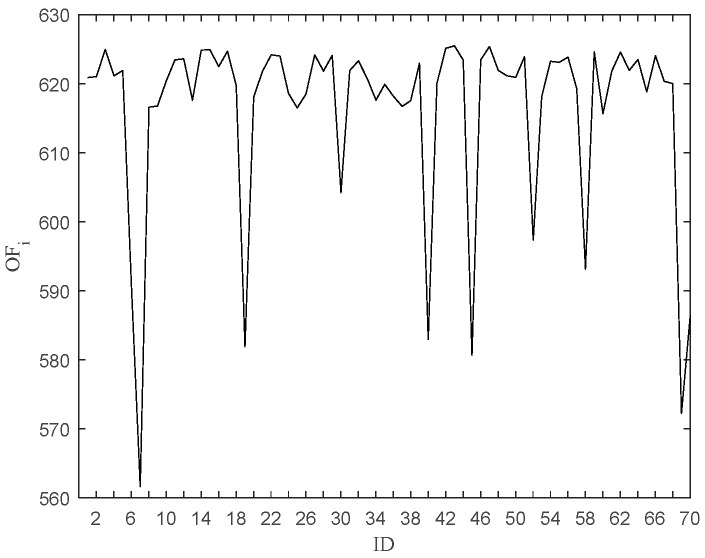
OF_i_ after removing the ith point.

**Figure 7 entropy-27-00582-f007:**
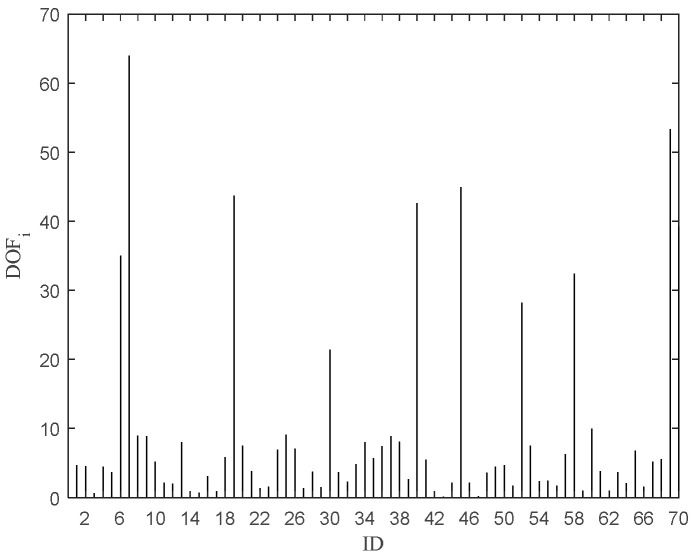
DOF_i_ after removing the ith point.

**Figure 8 entropy-27-00582-f008:**
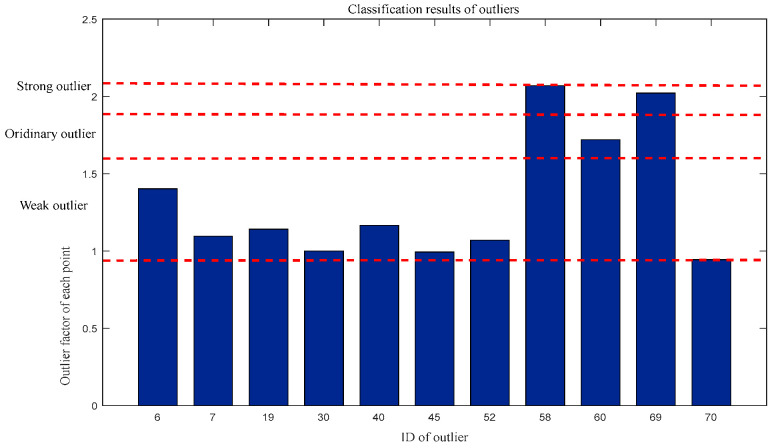
Classification of outliers by outlier factor.

**Figure 9 entropy-27-00582-f009:**
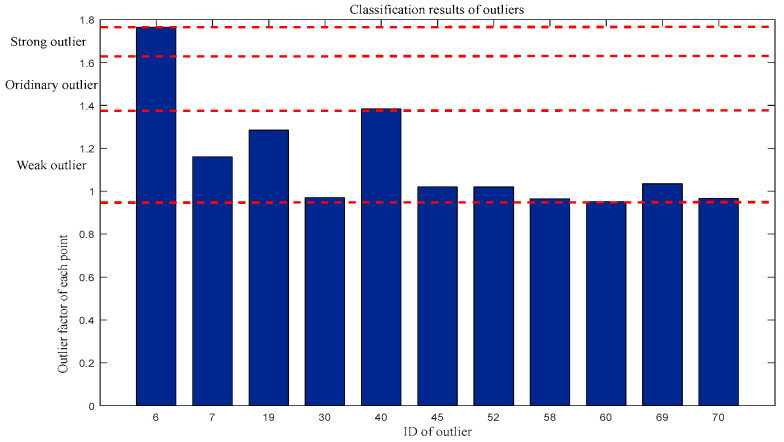
Classification of outliers in terms of first-dimensional outlier factors.

**Figure 10 entropy-27-00582-f010:**
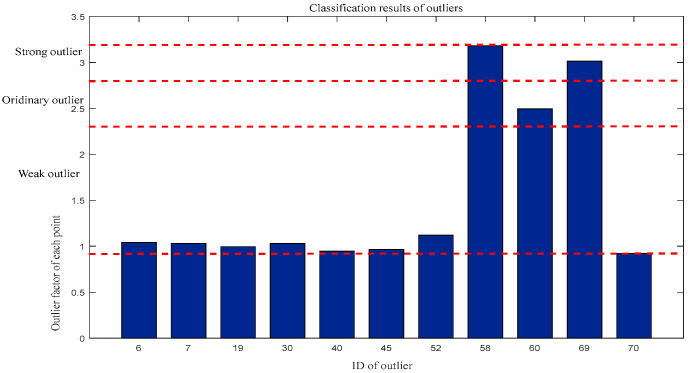
Classification of outliers in terms of second-dimensional outlier factors.

**Table 1 entropy-27-00582-t001:** Comparison of value reduction in the objective function.

ID	OFi	DOFi	ID	OFi	DOFi
1	620.861	4.726	36	618.171	7.416
2	621.007	4.579	37	616.718	8.869
3	624.956	0.631	38	617.530	8.057
4	621.119	4.468	39	622.921	2.667
5	621.859	3.692	40	582.946	42.641
6	590.532	35.055	41	620.065	5.522
7	561.651	63.936	42	625.105	0.482
8	616.612	8.975	43	625.460	0.127
9	616.737	8.851	44	623.401	2.187
10	620.375	5.213	45	580.656	44.931
11	623.438	2.149	46	623.428	2.159
12	623.586	2.001	47	625.368	0.220
13	617.592	7.995	48	621.934	3.653
14	624.872	0.715	49	621.112	4.475
15	624.898	0.688	50	620.913	4.674
16	622.478	3.109	51	621.922	1.698
17	624.673	0.915	52	597.338	28.250
18	619.707	5.881	53	618.114	7.473
19	581.889	43.698	54	623.217	2.370
20	618.095	7.492	55	623.097	2.490
21	621.764	3.823	56	623.842	1.745
22	624.180	1.407	57	619.280	6.307
23	623.996	1.590	58	593.137	32.451
24	618.602	6.985	59	624.614	0.973
25	616.488	9.099	60	615.654	9.933
26	618.493	7.094	61	621.773	3.815
27	624.182	1.405	62	624.578	1.009
28	621.800	3.787	63	621.914	3.673
29	624.103	1.484	64	623.464	2.124
30	604.217	21.370	65	618.801	6.787
31	621.894	3.693	66	624.016	1.571
32	623.293	2.295	67	620.344	5.243
33	620.711	4.876	68	620.007	5.581
34	617.619	7.969	69	572.235	53.352
35	619.891	5.696	70	586.382	39.205

**Table 2 entropy-27-00582-t002:** The points in the D1 outlier candidate set and each one-dimensional outlier factor (Outlier factors in descending order).

ID	LOF	LOF1	LOF2
58	2.069	0.965	3.180
69	2.021	1.034	3.013
60	1.720	0.951	2.495
6	1.404	1.764	1.041
40	1.166	1.384	0.948
19	1.141	1.284	0.997
7	1.096	1.160	1.031
52	1.070	1.019	1.121
30	1.000	0.970	1.031
45	0.993	1.021	0.966
70	0.944	0.966	0.922

**Table 3 entropy-27-00582-t003:** Comparison of D1 clustering indicators.

Dataset	CH Index	Dunn Index	I Index	S Index
Original dataset D1	65.4948	0.0431	41.4020	120.4470
Dataset with outliers removed	103.3029	0.0966	52.5155	131.2707

**Table 4 entropy-27-00582-t004:** Reachable densities for each point and one dimension in the outlier candidate set.

ID	lrd_k_	lrd_k1_	lrd_k2_
58	0.070	0.184	0.077
69	0.075	0.166	0.081
60	0.085	0.187	0.097
6	0.071	0.089	0.354
40	0.084	0.110	0.398
19	0.084	0.118	0.382
7	0.102	0.151	0.357
52	0.099	0.168	0.332
30	0.092	0.184	0.357
45	0.104	0.168	0.388
70	0.093	0.185	0.407

**Table 5 entropy-27-00582-t005:** UCI datasets.

Dataset	Data Volume	Number of Attributes	Number of Classifications
Iris	150	4	3
Wine	178	13	3
Yeast	1484	8	10
UKM	403	5	4
Seeds	210	7	3
Wdbc	569	30	2
Speech	3686	400	4

**Table 6 entropy-27-00582-t006:** Comparison of pruning accuracy.

Dataset	FCM	DBSCAN	PMLDOF
Iris	1	0.85	0.9
Wine	0.9	0.90	0.8
Yeast	1	0.90	1
UKM	1	1	0.9
Seeds	0.9	0.8	0.8
Wdbc	1	0.9	0.9
Speech	1	0.8	0.9

**Table 7 entropy-27-00582-t007:** Comparison of the remaining data after pruning.

Dataset	FCM	DBSCAN	PMLDOF
Iris	29	35	32
Wine	47	52	45
Yeast	472	530	497
UKM	81	90	88
Seeds	54	68	63
Wdbc	102	125	118
Speech	1016	1424	1386

**Table 8 entropy-27-00582-t008:** Comparison of experimental results on the UCI datasets.

	FOLOF					LOF				
Dataset	TP	FP	Pr	Nf	t	TP	FP	Pr	Nf	t
Iris	9	1	0.900	0.100	6.586	6	4	0.600	0.400	15.182
Wine	6	2	0.750	0.250	55.221	5	3	0.625	0.375	290.532
Yeast	18	2	0.900	0.100	387.552	15	5	0.750	0.250	1318.221
UKM	4	1	0.800	0.200	187.462	4	1	0.800	0.200	562.113
Seeds	9	1	0.900	0.100	61.325	7	3	0.700	0.300	310.234
Wdbc	4	1	0.800	0.200	213.256	3	2	0.600	0.400	627.453
Speech	8	2	0.800	0.200	965.523	7	3	0.700	0.300	2563.278

**Table 9 entropy-27-00582-t009:** Comparison of clustering effects before and after removal of outliers for the four datasets.

	Complete Dataset				Dataset After Removing Outliers			
Dataset	CH Index	Dunn Index	I Index	S Index	CH Index	Dunn Index	I Index	S Index
Iris	512.642	0.451	21.203	124.695	548.865	0.604	20.930	101.740
Wine	50.390	0.214	4.963	144.688	50.647	0.218	5.295	122.652
Yeast	194.378	0.024	0.013	2.759	189.394	0.025	0.013	2.862
UKM	92.189	0.069	0.076	6.226	92.560	0.069	0.079	6.190
Seeds	155.975	0.059	9.097	5.362	172.956	0.067	7.964	5.763
Wdbc	232.061	0.024	4.298	4.535	276.7681	276.768	4.328	4.856
Speech	1698.421	0.017	4.597	6.458	1856.015	0.033	4.683	6.854

**Table 10 entropy-27-00582-t010:** AUC experimental results on 7 UCI datasets.

Datasets	FOLOF	LOF	IFOREST	BLDOD	PEHS
Iris	0.9545	0.7873	0.9256	0.9377	0.9432
Wine	0.8978	0.7781	0.8801	0.8356	0.8618
Yeast	0.9896	0.8532	0.9254	0.9156	0.9635
UKM	0.8789	0.8511	0.8412	0.8612	0.8563
Seeds	0.9654	0.8735	0.9463	0.9487	0.9502
Wdbc	0.8951	0.8423	0.8856	0.8651	0.8752
Speech	0.9372	0.8947	0.9284	0.9301	0.9221

**Table 11 entropy-27-00582-t011:** Wilcoxon rank sum test results on UCI datasets.

Methods	R+	R−	*p*-Value	Assumption
FOLOF vs. LOF	28	0	0.0156	reject
FOLOF vs. IFOREST	28	0	0.0156	reject
FOLOF vs. BLDOD	28	0	0.0156	reject
FOLOF vs. PEHS	28	0	0.0156	reject

**Table 12 entropy-27-00582-t012:** Classification of outliers in the UCI datasets.

Datasets	Weak Outliers	Ordinary Outliers	Strong Outliers
Iris	103, 105, 108, 109, 110	101, 104, 106	107
Wine	131, 132, 133, 134	138	136
Yeast	1117, 1121, 1123, 1124, 1125, 1127, 1128	1119, 1122, 1126, 1133, 1134, 1135	1123, 1129, 1130, 1131, 1132
UKM	226, 227	225	229
Seeds	141, 143,145,146, 149,150	142,	147, 148
Wdbc	572, 573	574	570
Speech	3689, 3694, 3695	3687, 3690, 3696	3691, 3692

**Table 13 entropy-27-00582-t013:** Outlier factors and dimension outlier factors for the top 10 outliers.

ID	LOF	LOF1	LOF2	LOF3	LOF4	LOF5
6	1.779	1.331	4.512	1.057	1.000	1.061
16	1.636	1.021	1.113	3.862	1.000	1.100
5	1.605	1.530	1.030	3.348	1.000	1.061
21	1.602	0.952	4.105	0.988	1.000	1.111
79	1.489	1.431	1.545	1.099	2.429	0.961
146	1.485	1.051	0.952	3.076	0.958	1.330
17	1.457	0.983	3.292	0.967	1.000	1.038
1	1.436	2.664	1.112	1.396	1.000	1.028
3	1.424	1.805	1.767	1.510	1.029	1.061
55	1.400	1.403	1.946	1.601	1.000	1.040

**Table 14 entropy-27-00582-t014:** Outlier classification of representative players.

Local Outlier Factors	Weak Outliers	Ordinary Outliers	Strong Outliers
LOF_1_	6, 16, 17, 21, 55, 146	5, 79, 3	1
LOF_2_	1, 3, 5, 16, 55, 79, 146	17	6, 21
LOF_3_	1, 3, 6, 17, 21, 79, 55	-	5, 16, 146
LOF_4_	1, 3, 5, 6, 16, 17, 21, 55, 146	-	79
LOF_5_	1, 3, 5, 6, 17, 55, 79	16, 21	146

## Data Availability

The original contributions presented in this study are included in the article. Further inquiries can be directed to the corresponding author. Data will be made available on request.

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
