# Peer review of "Outlier Detection and Explanation Method Based on FOLOF Algorithm"

_entropy, 2025, doi:10.3390/e27060582_

Round 1

Reviewer 1 Report

Comments and Suggestions for Authors

Improve figures: the text in many figures is not visible. Consider enhancing their quality to ensure readability.

Improve equations and how they are linked to each other or to the text. It is currently difficult for a reader to follow the reasoning and understand how the equations relate to the narrative.

Improve consistency in how variables are mentioned in the text. For example, variables like P₁ are always shown as superscript; the formatting should be consistent throughout the manuscript.

In Equation 7, verify whether the index of the second summation is i or if it should rather be j.

The method section can be improved by adding a paragraph between Sections 3 and 3.1 to introduce the subsections and provide context before detailing each one.

The conclusion section is currently a summary of the paper. Consider rewriting it to clearly present the main findings and their implications.

The limitations of the study have not been clearly stated; consider explicitly addressing them.

The manuscript structure and flow could benefit from improvements such as reorganization or additional subheadings, although this is not strictly required.

Comments on the Quality of English Language

The manuscript would benefit from language editing. Writing is not always in academic style and needs improvement.

Author Response

     First of all, we would like to express our sincere gratitude to the editors and anonymous reviewers for their time, efforts and recognition given to our manuscript entitled “Outlier Detection and Explanation Method based on FOLOF Algorithm”. 

     Secondly, it is worth pointing out that the editors’ and reviewers’ comments and suggestions have significantly helped us improve the quality and presentation of our manuscript further. In response to their valuable and insightful feedback, we have meticulously revised the paper, addressing each comment individually, with the main revisions highlighted in blue in the revised manuscript and listed as follows.

Reviewer 2 Report

Comments and Suggestions for Authors

The authors have made great improvements on the revised manuscript. However, there are still some issues to be addressed.

1.The experimental dataset can include diverse fields such as high-dimensional vibration time-series fault datasets.

2.While the manuscript proposes an outlier detection method based on FOLOF algorithm, it mainly focuses on the traditional techniques. It is advised to include the latest works, such as deep learning techniques in outlier detection, especially advanced explanation techniques like match pursuit network.

3.The limitations of the proposed method could be further added.

4.Some figures in the manuscript are relatively simple and can be improved.

Author Response

(The authors gave the same response as above.)

Round 2

Reviewer 1 Report

Comments and Suggestions for Authors

The authors have addressed the comments.

Author Response

We greatly appreciate your comments and hard work on our manuscript.

Reviewer 2 Report

Comments and Suggestions for Authors

The revised mansucript has made some improvements. But some issues are still not fully addressed.

  1. Beyond traditional clustering and LOF-based explainable methods, emerging explainable learning methods may offer new insights for outlier explanation. However, the paper does not cover these contents. It is recommended that the author pay attention to and summarize the latest developments in this field to further enhance the literature review.
  2. The paper only emphasizes the differences of the proposed method from traditional clustering and LOF-based algorithms, but fails to adequately review the latest deep learning and explainable learning-related works. For instance, representative deep learning methods like Matching Pursuit Network for Mechanical Fault are not mentioned.
  3. Although the paper proposes an outlier explanation method based on the Golden Section method, the accuracy and effectiveness of this explanation method are not sufficiently verified.

Author Response

First of all, we would like to express our sincere gratitude to the editors and anonymous reviewers for their time, efforts and recognition given to our manuscript entitled “Outlier Detection and Explanation Method based on FOLOF Algorithm”. Secondly, it is worth pointing out that the editors’ and reviewers’ comments and suggestions have significantly helped us improve the quality and presentation of our manuscript further. In response to their valuable and insightful feedback, we have meticulously revised the paper, addressing each comment individually, with the main revisions highlighted in yellow in the revised manuscript and listed as follows.

Response to Comments of Reviewer 

Comment 1: Beyond traditional clustering and LOF-based explainable methods, emerging explainable learning methods may offer new insights for outlier explanation. However, the paper does not cover these contents. It is recommended that the author pay attention to and summarize the latest developments in this field to further enhance the literature review.

Response: We sincerely appreciate the valuable comments. We have supplemented the content and collected some references, which are as follows:

In recent research, several emerging explanation methods have provided novel insights for outlier detection,including LIME-AD[15], DTOR[16], and ProtoPNets[17].

[15] Hariharan, S., Jerusha, Y. A., Suganeshwari, G., Ibrahim, S. S., Tupakula, U., & Varadharajan, V. (2025). A Hybrid Deep Learning Model for Network Intrusion Detection System using Seq2Seq and ConvLSTM-Subnets. IEEE Access.

[16]Crupi, R., Regoli, D., Sabatino, A. D., Marano, I., Brinis, M., Albertazzi, L., ... & Cosentini, A. C. (2024). DTOR: Decision Tree Outlier Regressor to explain anomalies. arXiv preprint arXiv:2403.10903.

[17]Willard, F., Moffett, L., Mokel, E., Donnelly, J., Guo, S., Yang, J., ... & Rudin, C. (2024). This looks better than that: Better interpretable models with protopnext. arXiv preprint arXiv:2406.14675.

Comment 2:The paper only emphasizes the differences of the proposed method from traditional clustering and LOF-based algorithms, but fails to adequately review the latest deep learning and explainable learning-related works. For instance, representative deep learning methods like Matching Pursuit Network for Mechanical Fault are not mentioned.

Response: We sincerely appreciate the valuable comments. We have supplemented the content and collected some references, which are as follows:

In the latest research,Bhurtyal S et al.[28] presents a deep learning architecture exploiting a features choice module and mask generation module in order to learn both components of explanations. The extensive experimental campaign carried out on synthetic and real data sets provides empirical evidence demonstrating the quality of the results returned by M2OE for explaining single outliers and M2OE-groups for explaining outlier groups. Papastefanopoulos V et al.[29] introduces an interpretable approach to unsupervised outlier detection by combining normalizing flows and decision trees. Posthoc statistical significance testing demonstrated that interpretability in unsupervised outlier detection can be achieved without significantly compromising performance, making it a valuable option for applications that require transparent and understandable anomaly detection. Angiulli F et al.[30] presents a deep learning architecture exploiting a features choice module and mask generation module in order to learn both components of explanations.Experiments were conducted on both artificial and real datasets, and compared with competitors to validate the effectiveness of the proposed method.

[28]Bhurtyal, S., Bui, H., Hernandez, S., Eksioglu, S., Asborno, M., Mitchell, K. N., & Kress, M. (2025). Prediction of waterborne freight activity with Automatic identification System using Machine learning. Computers & Industrial Engineering, 200, 110757.

[29]Papastefanopoulos, V., Linardatos, P., & Kotsiantis, S. (2025). Combining normalizing flows with decision trees for interpretable unsupervised outlier detection. Engineering Applications of Artificial Intelligence, 141, 109770.

[30]Angiulli, F., Fassetti, F., Nisticò, S., & Palopoli, L. (2024). Explaining outliers and anomalous groups via subspace density contrastive loss. Machine Learning, 113(10), 7565-7589.

Comment 3:Although the paper proposes an outlier explanation method based on the Golden Section method, the accuracy and effectiveness of this explanation method are not sufficiently verified.

Response: We sincerely appreciate the valuable comments. The accuracy and effectiveness of the golden-section-based outlier interpretation method are fully demonstrated in the experiments of Section 6 of the paper. We have also supplemented corresponding explanations in the Conclusion section.
